# A Balance between Transmembrane-Mediated ER/Golgi Retention and Forward Trafficking Signals in Glycophorin-Anion Exchanger-1 Interaction

**DOI:** 10.3390/cells11213512

**Published:** 2022-11-06

**Authors:** Kate Hsu, Ting-Ying Lee, Jian-Yi Lin, Pin-Lung Chen

**Affiliations:** 1The Laboratory of Immunogenetics, Department of Medical Research, MacKay Memorial Hospital, Tamsui, New Taipei City 251020, Taiwan; 2MacKay Junior College of Medicine, Nursing, and Management, New Taipei City 25245, Taiwan; 3Institute of Biomedical Sciences, MacKay Medical College, New Taipei City 25245, Taiwan; 4Department of Exercise & Health Sciences, University of Taipei, Taipei 100234, Taiwan

**Keywords:** red blood cells (RBCs; erythrocytes), anion exchanger-1 (AE1; band 3; SLC4A1), glycophorin A (GPA), glycophorin B (GPB), GP.Mur (Miltenberger subtype III; Mi.III), ER/Golgi retention, oligomerization, transmembrane domain (TMD), trafficking, membrane protein

## Abstract

Anion exchanger-1 (AE1) is the main erythroid Cl^−^/HCO_3_^−^ transporter that supports CO_2_ transport. Glycophorin A (GPA), a component of the AE1 complexes, facilitates AE1 expression and anion transport, but Glycophorin B (GPB) does not. Here, we dissected the structural components of GPA/GPB involved in glycophorin-AE1 trafficking by comparing them with three GPB variants—GPBhead (lacking the transmembrane domain [TMD]), GPBtail (mainly the TMD), and GP.Mur (glycophorin B-A-B hybrid). GPB-derived GP.Mur bears an O-glycopeptide that encompasses the R18 epitope, which is present in GPA but not GPB. By flow cytometry, AE1 expression in the control erythrocytes increased with the GPA-R18 expression; *GYP.Mur*^*+*/*+*^ erythrocytes bearing both GP.Mur and GPA expressed more R18 epitopes and more AE1 proteins. In contrast, heterologously expressed GPBtail and GPB were predominantly localized in the Golgi apparatus of HEK-293 cells, whereas GBhead was diffuse throughout the cytosol, suggesting that glycophorin transmembrane encoded an ER/Golgi retention signal. AE1 coexpression could reduce the ER/Golgi retention of GPB, but not of GPBtail or GPBhead. Thus, there are forward-trafficking and transmembrane-driven ER/Golgi retention signals encoded in the glycophorin sequences. How the balance between these opposite trafficking signals could affect glycophorin sorting into AE1 complexes and influence erythroid anion transport remains to be explored.

## 1. Introduction

Anion exchanger-1 (AE1; band 3; SLC4A1) and Glycophorin A (GPA) are the two most abundant membrane proteins in human red blood cells (RBCs), each with ~10^6^ molecules per cell [1,2]. AE1 is a large transmembrane protein (911 amino acids) with multiple functions that correspond to its different structural domains. The transmembrane domain of AE1 functions as a Cl^−^/HCO_3_^−^ transporter that could “ping-pong” or exchange one anion for another in and out of the red cells [3,4]. Because most blood CO_2_ is carried in the form of Cl^−^/HCO_3_^−^, AE1-mediated Cl^−^/HCO_3_^−^ transport across the erythrocyte membrane is important for physiologic CO_2_ respiration. Cl^−^/HCO_3_^−^ transport by AE1 also helps stabilize cellular pH [5]. The N-terminal cytoplasmic domain of AE1, interacts with major submembranous proteins (i.e., ankyrin, protein 4.2, adducin, and 4.1R) and forms physical linkages between the erythrocyte membrane and the underneath spectrin-actin junctional complexes [6,7,8,9,10]. The latter function of AE1 provides mechanical support to maintain the morphology and membrane integrity of the erythrocytes.

The surface expression of AE1 is greatly facilitated by GPA [11,12,13]. GPA, bearing a single transmembrane span, is extensively glycosylated with one N-linked and 15 O-linked glycans, as well as sialic acid moieties [14,15]. The sialic acid moieties of GPA contribute to the negatively charged glycocalyx on the erythrocyte surface, which prevents cell–cell adhesion during circulation. The knockout of glycophorin A in mice reduces the expression of O-linked sialoglycoproteins on the erythrocyte membrane, resulting in RBC hypersensitivity to osmotic pressure [16,17]. Glycophorin B (GPB), a homologue of GPA with ~95% nucleotide identity, is also heavily O-glycosylated and decorated with sialic acid moieties [18,19]; it presumably also provides mechanical support for erythrocyte membrane stability.

Numerous studies have revealed a structural–functional association between AE1 and GPA [20]. Early fluorescence photobleaching recovery and polarized fluorescence depletion experiments found that AE1 and GPA in the sickle and normal RBCs aggregate and exhibit similar mobility responses to osmotic stimulation [21]. AE1 and GPA begin to interact in the endoplasmic reticulum (ER) where their protein expression levels are tightly coupled [22,23]. The discovery of the Wright b (Wr^b^) antigen on the interface between AE1 and GPA is solid proof that the two proteins physically interact on the RBC membrane [24,25,26]. Their tightly coupled expression levels have physiological and pathophysiological implications. For example, selective knockout of erythroid AE1 results in severe spherocytosis and hemolytic anemia as well as complete deficiency of GPA on the erythrocyte membrane [22,27,28]. Intriguingly, *GYPA* mRNA could still be found in the erythroid precursor cells of these AE1 knockout mice, indicating that the expression of erythroid GPA protein requires AE1 [22]. On the other hand, *GYPA* knockdown by shRNA in human erythroleukemic K562 cells does not affect the surface expression of AE1 [23], indicating that AE1 is capable of trafficking to the plasma membrane by itself. However, in the RBCs that lack both GPA and GPB (the rare M^k^M^k^ blood type), the anion transport activity of AE1 is significantly reduced, though the protein content of AE1 is not significantly affected [29]. Thus, even though AE1 protein expression does not require GPA [22,29], GPA interaction with AE1 could support the normal anion transport function of AE1 and optimize AE1 surface expression [11,29]. For comparison, though GPB is also abundantly expressed in the erythrocyte membrane (200,000 or one-fifth the number of GPA molecules per erythrocyte), GPB does not enhance AE1 surface expression or AE1-mediated Cl^−^ flux [12,13,29,30,31]. Yet GPB is a component of AE1-associated complexes on the RBC membrane [11,32].

This study aimed to dissect the glycophorin structural components in their abilities to facilitate glycophorin-AE1 trafficking and surface expression. By comparing GPA/GPB with three naturally occurred *GYPB* mRNA variants cloned from the reticulocytes of healthy people (Figure 1A: GP.Mur, GPBhead and GPBtail), we identified an ER/Golgi retention signal encoded in the glycophorin transmembrane domain. The forward-trafficking signals (i.e., the R18 epitope present in both GPA and GP.Mur) could override ER/Golgi retention by the glycophorin transmembrane domain. Heterologous coexpression of AE1 could also counteract such ER/Golgi retention, suggesting that trafficking of AE1 and glycophorin A/B and variants also depends on their intricate interaction, modification, and sorting into the large AE1 complexes.

## 2. Materials and Methods

### 2.1. Red Blood Cell Samples and Cultured Cells

The Mackay Memorial Hospital Institutional Review Board approved the collection of human blood from consented non-diseased donors (MMH-IRB registration: MMH-I-S-517; 10MMHIS096). The Miltenberger subtype III (Mi.III) phenotype was serologically screened with anti-Mi^a^, anti-Mur, and anti-Hil antisera. Homozygous GP.Mur (Mi.III^+/+^) RBC samples were differentiated from heterozygous ones by their lack of GPB expression on Western blot. HEK-293 cells were maintained in Dulbecco’s Modified Eagle’s Medium (DMEM) containing 10% fetal bovine serum (FBS) and penicillin-streptomycin.

### 2.2. Cloning of Glycophorin Transcripts

The total RNA was extracted from the reticulocytes of healthy donors using the QIAamp RNA Blood Kit (Qiagen, Hilden, Germany) and then reverse transcribed into cDNA. Glycophorin B transcripts were PCR-amplified using the following primer pair: 5′-AAGCTTTTTGCACTAACTTCAGGAACCAGC (GYPA19HindIII) and 5′-GGCATAAGCAAAGGAATAGCAGG (GYPB453r) with PfuUltra High-fidelity DNA Polymerase AD (Stratagene). The PCR products were cloned into TOPO vectors using a Zero Blunt TOPO PCR Cloning Kit for Sequencing (Invitrogen). Individual *GYPB* clones were verified by DNA sequencing. An alternatively spliced variant of glycophorin B (GenBank accession EU338230) lacks *GYPB* exon 5 that encodes the TMD; this glycophorin B spliced variant was thus named “GPBhead”. Another *GYPB* spliced variant lacks exons II-IV of *GYPB* and encompasses only exons I, V (coding for the TMD), and VI, and was thus named “GPBtail” (Figure 1A).

*GYPB* cDNA (NM002100) and its naturally occurred spliced variants—*GYPBhead* (EU338230) and *GYPBtail* (EU338238)—were each subcloned into pcNDA3.1 (Invitrogen, Waltham, MA, USA) to generate pcGPB, pcGPBhead, and pcGPBtail, respectively. For confocal imaging experiments, *GYPB*, *GYP.Mur*, *GYPBhead*, and *GYPBtail* were each subcloned into pEGFP-N2 (Clontech, Mountain View, CA, USA) to generate the green fluorescence fusion protein constructs—pGPBgfp, pGPMURgfp, pGPBhead-GFP, and pGPBtail-GFP. *GYPA* cDNA (NM002099) was subcloned into pEYFP-N1 (Clontech) to generate a yellow fluorescence fusion construct—pGPAyfp. The other plasmids used in this study (pcGPA, pcGP.Mur, pcAE1, and bicistronic pCIG-AE1) were described previously [11].

### 2.3. Transfection

Heterologous gene expression in HEK-293 cells was introduced by transient transfection using Lipofectamine 2000 Transfection Reagent (Invitrogen) or T-Pro Non-liposome Transfection Reagent II (T-Pro Biotechnology, New Taipei City, Taiwan). AE1 and one of the glycophorin plasmids were either singly transfected or co-transfected at equimolar ratios or at other molar ratios as specified into HEK-293 cells cultured in 6-well plates. For all experimental sets, the control or vector plasmids (e.g., pcDNA3.1) were supplemented to make the total amount of DNA for transfection constant. All imaging experiments were conducted during 36–72 h post-transfection.

### 2.4. Flow Cytometry and Monoclonal Antibodies (mAb)

To assess the surface expression of AE1, human red blood cells were immunolabelled with BRIC 71 and BRIC 6 (International Blood Group Reference Laboratory [IBGRL]), two monoclonal antibodies that target the extracellular loops of AE1 [34,35]. BRIC 170 recognizes the N-terminal cytoplasmic domain of AE1 (amino acids 368–382). R18 (IBGRL), a monoclonal antibody that targets an extracellular region common to GPA and GP.Mur (its epitope corresponding to residues 39–56 or residues 49–52 of GPA) [35,36,37], was used to quantify the surface levels of GPA in control (non-Mi.III) RBCs (or the combined levels of GPA and GP.Mur in Mi.III RBCs). After primary antibody labeling, the cells were stained with Alexa Fluor 660-conjugated anti-mouse antibody (Invitrogen). In flow cytometric measurements, surface protein expression levels were assessed using the FL4-H channel of FACSCalibur (BD).

### 2.5. Confocal Microscopy

To study how the different glycophorin structural components could affect AE1-glycophorin trafficking and surface expression, HEK-293 cells were transiently transfected with AE1 alone or together with pcGPA/pGPAyfp, pGPBgfp, pGPMURgfp, pGPBtail-GFP, or pGPBhead-GFP. As the spectra of YFP and GFP were quite similar (both showed green fluorescence in our imaging system), we avoided co-transfection of GPAyfp and GPBgfp (or other GPBgfp variants). For the triple expression of AE1, GPA, and GPBgfp, on day 2 or 3 post-transfection, the triply transfected cells were fixed and permeabilized, and then immunostained with Alexa Fluor 568-conjugated anti-GPA BRIC163, and mouse anti-AE1 antibodies (BRIC 6 + BRIC 71 + BRIC 170) which were then labeled with Alexa Fluor 647-conjugated anti-mouse antibody. Confocal imaging utilized a Leica TCS SP equipped with an argon/Krypton laser. To determine subcellular localization, the ER-Tracker blue-white DPX dye (Invitrogen), BODIPY TR-ceramide (Invitrogen), and PE-Cy5 mouse anti-human CD107a antibody (BD Pharmingen, San Diego, CA, USA) were used to label the endoplasmic reticulum (ER), the Golgi apparatus, and lysosomes, respectively.

## 3. Results

### 3.1. The Expression Levels of AE1 and the Glycophorin R18 Epitope were Quantitatively Directly Correlated on the Human Erythrocyte Membrane

From sequence alignment of GPA, GPB and the B-A-B variant GP.Mur (Figure 1), GPB lacks an extracellular, O-glycosylated domain that is present in both GP and GP.Mur. Since GPB does not support AE1 surface expression as GPA or GP.Mur, we examined whether this O-glycopeptide absent from GPB might play a role in enhancing AE1 expression. We used the R18 monoclonal antibody that targets this O-glycopeptide in residues 39–56 or residues 49–52 of GPA [35,36,37]. The R18 epitope is also present in GP.Mur protein, as *GYP.Mur* was derived from the insertion of *GYPA* in the middle of *GYPB* [38]. We found that the expression levels of AE1 and the R18 epitope on GPA were directly correlated in human RBC samples (Figure 2: Pearson correlations = 0.383; significance = 0.0532), suggesting a role of the R18 epitope in supporting erythroid AE1 expression. This correlation was not strong, which could be due to that AE1 could traffic to the cell surface without the chaperone activity of GPA [13,29]. Notably, the RBC samples from *GYP.Mur^+/+^* subjects (carrying homologous *GYP.Mur* alleles that completely replace the two *GYPB* alleles) exhibited more R18 epitopes and AE1 on their cell membrane, compared to the non-Miltenberger samples (Figure 2). *GYP.Mur^+/+^* and the control (non-Miltenberger) RBCs exhibit similar levels of GPA [11]; thus, the higher R18 levels in *GYP.Mur^+/+^* RBCs were attributed to the R18 epitope from GP.Mur protein. The direct quantitative correlation between expressions of the R18 epitope and AE1 (Figure 2) therefore indicated that the R18 epitope of GPA/GP.Mur could support the forward trafficking of AE1 protein complexes to the erythrocyte surface.

### 3.2. Differential Subcellular Localization Patterns between AE1 and GPA/GPB/GP.Mur

Since mature RBCs lack most intracellular organelles, we next studied glycophorin-AE1 trafficking in a heterologous expression system. The transfection efficiencies using erythroleukemic cell lines were generally very low and inconsistent, so for the following experiments, we used the human embryonic kidney HEK-293 cell line which could yield stable and high-transfection efficiencies (30–60%).

To explore how GPA/GPB/GP.Mur could influence AE1 trafficking or vice versa, we first verified the previous observation that AE1 is capable of forward trafficking to the plasma membrane by itself (Figure 3A, left). Due to the high degree of sequence homology among GPA, GPB, and GP.Mur (Figure 1), it was difficult to find an antibody for immunolabeling that recognizes only GPB and not GPA or GP.Mur, or an antibody that recognizes GP.Mur and not GPA or GPB. Therefore, we tagged GFP to the C-termini of GPB and GP.Mur (following the TMD) as fluorescence trackers. We found that GPA and GPMURgfp each expressed substantially on the plasma membrane as well as in the organelle membrane, while GPBgfp primarily aggregated intracellularly (Figure 3A).

Co-transfection of equimolar AE1 (blue) and glycophorin A (red) in HEK-293 cells resulted in nearly complete colocalization (purple) in the intracellular and surface membranes (Figure 3B). For comparison, GPMURgfp (green) and AE1 (red) were also colocalized intracellularly and on the plasma membranes, but they did not show as complete colocalization as that between GPAyfp and AE1 (Figure 3B,C). Intriguingly, in contrast to the large intracellular aggregates of GPBgfp expression alone (Figure 3A middle), AE1 co-transfection reduced the sizes of GPBgfp aggregates or blobs. AE1 and GPBgfp were colocalized on the plasma membrane but not in the intracellular organelles (Figure 3D). AE1 coexpression thus appeared to redirect the intracellular trafficking of GPB and to promote GPB surface expression.

The ratio of the protein copy number of GPA: GPB: AE1 in human erythrocytes is 1:0.2:1. We mimicked their erythroid gene expression heterologously in HEK-293 cells by cotransfecting the same molar ratio of the GPA, GPBgfp, and AE1 plasmids. We found that GPBgfp coexpression did not affect the nearly perfect colocalization between GPA and AE1 (Figure 3E). GPBgfp was partially colocalized with the GPA-AE1 complexes on the plasma membrane, despite that its cytosolic expression remained substantial (Figure 3E).

### 3.3. Substantial Localization of GPB in the ER and Golgi Apparatus Could Be Redirected to the Plasma Membrane upon AE1 Coexpression

To verify the sites where GPB aggregated (Figure 3), we labeled the transfected cells with a fluorescent ER tracker. Figure 4A showed the single expression of each glycophorin fusion protein—GPAyfp, GPMURgfp, and GPBgfp, respectively. While GPAyfp and GPMURgfp were expressed on the plasma membrane, GPBgfp showed little surface expression but substantial aggregation in the ER. The degree of GPMURgfp localization to the ER was between that of GPBgfp and GPAyfp. AE1 coexpression did not affect much of the subcellular localization of GPAyfp and GPMURgfp (Figure 4B,C); on the other hand, AE1 coexpression with GPBgfp dissipated some intracellular aggregates of GPBgfp (Figure 4D).

We next labeled GPBgfp-transfected cells with BODIPY TR-ceramide, a red fluorescent marker for Golgi complexes. We found that the sites of massive GPBgfp aggregation also included the Golgi apparatus (Figure 5). In addition, we found that most GPB-GFP aggregates were not targeted to degradation in the lysosomes, as the merged yellow color from colocalization of the lysosome marker anti-LAMP/AF568 and GPBgfp was barely seen (Figure 5). For comparison, the GPMURgfp fusion protein expressed on both intracellular and plasma membranes, and did not aggregate in the Golgi apparatus or the ER (Figure 4C and Figure 5A,B). Since GPMURgfp differs from GPBgfp only by an extra 31 amino acids that encompass the Mur and the R18 epitopes and GPA lacks the Mur antigen (Figure 1), the R18 epitope thus encodes a forward-trafficking signal that could counteract ER/Golgi retention of GPBgfp (Figure 4 and Figure 5).

### 3.4. The Glycophorin Transmembrane Domain Encodes a Golgi Retention Signal

To probe into the nature of GPBgfp aggregation, we utilized two naturally occurring, exon-skipped variants of *GYPB*—*GYPBhead* and *GYPBtail*, cloned from the reticulocytes of healthy subjects. To track their cellular expression, we also tagged GFP to their C-termini to make GPBhead-GFP and GPBtail-GFP fusion proteins. The N-terminal sequence of GPBhead is identical to that of GPB. On the other hand, GPBhead lacks exon 5 in *GYPA*/*GYPB* that codes for the transmembrane sequence and was predicted to be novel and “non-transmembrane” (either cytosolic or membrane-associated) by TMHMM 2.0, a structural prediction program for transmembrane helices [39]. In contrast, GPBtail lacks most of the N-terminal, glycosylated domain but retains the transmembrane region (Figure 1).

By tracking their subcellular localization, we found that, unlike GPBgfp, the GPBhead-GFP fusion protein expressed almost everywhere in the cytoplasm (Figure 5C). On the other hand, similar to the GPBgfp fusion protein, the GPBtail-GFP fusion was also concentrated in the Golgi apparatus (Figure 3A and Figure 5D). Thus, the glycophorin TMD encodes an ER/Golgi retention signal. While there was little GPBgfp targeted to the lysosomes, more GPBhead-GFP and GPBtail-GFP molecules were found in the lysosomes (Figure 5), hinting that these partial GPB structures (head or tail) might have failed the quality control step during membrane protein biosynthesis.

While AE1 coexpression could dissolve GPB aggregation in the ER and the Golgi, AE1 coexpression with either GPBtail or GPBhead did not affect their individual subcellular localization (Figure 6). Thus, the complete GPB sequence could be required for partitioning into AE1-associated protein complexes (Figure 3D).

## 4. Discussion

In this study, we identified that the glycophorin transmembrane domain encoded an ER/Golgi retention signal, which could be counteracted by forward-trafficking signals (i.e., the R18 epitope of GPA/GP.Mur; the cytoplasmic domain of GPA [33]) or by AE1 coexpression (Figure 7). GP.Mur lacks this cytoplasmic domain of GPA but still could promote AE1 surface expression (Figure 2) [11]. By structural deduction, the R18 epitope and its vicinity in GP.Mur and GPA thus encode a forward-trafficking signal (Figure 1). The protein structure of GPB differs from the structure of GPA by lacking (1) an extracellular O-glycopeptide that contains the R18 epitope, and (2) a large C-terminal, cytoplasmic domain (Figure 1). Thus, GPB was by and large an intracellular protein and did not support AE1 forward trafficking (Figure 1, Figure 3, Figure 4 and Figure 5). While GPBhead (lacking the transmembrane domain) was diffusive all over the cytosol and was not specifically targeted during intracellular trafficking, GPBtail (containing the TMD) aggregated in the ER and the Golgi apparatus like GPB (Figure 5 and Figure 6). As the TMD of glycophorin B is identical to the TMD of GP.Mur and is highly homologous to the TMD of GPA (Figure 1), the homologous glycophorin transmembrane region conceivably encompasses a ER/Golgi retention signal.

We did not observe any known amino acid motifs for ER or Golgi retention in the glycophorin alignment (Figure 1). On the other hand, Golgi retention could be driven by properties of the TMD that are associated with the propensity of membrane protein oligomerization (i.e., membrane protein partitioning into large AE1-associated complexes in this case) [41]. Intriguingly, in another study by Förster Resonance Energy Transfer by Fluorescence Lifetime Imaging (FLIM-FRET), AE1 and GPB do not interact directly (their molecular distance ≥ 10 nm and considered outside the range for dipole–dipole interaction) (unpublished data). However, GPB is sorted into AE1-associated protein complexes on the erythrocyte membrane [11,32]. Here, by heterologous expression, AE1 could direct GPB trafficking and drive GPB to the cell surface (Figure 3 and Figure 4). However, AE1 could not affect the expression of GPBtail or GPBhead (Figure 5 and Figure 6), indicating that the entire GPB structure is required for it to be recruited into AE1-associated complexes during protein sorting/partitioning, oligomerization, and post-translational modification/glycosylation in the ER/Golgi apparatus.

From FLIM-FRET measurements, AE1 and GPA proteins physically interact in the transfected cells (unpublished data). Though this study only observed AE1-GPA colocalization, the two proteins likely utilize the same trafficking route and begin oligomerization before they reach to the plasma membrane as AE1 protein complexes [11,24,25,26,32]. This idea is also supported by several lines of evidence. First, in the dual absence of GPA and GPB in M^k^M^k^ RBCs, the glycosylation profile of AE1 is altered [29]. Second, in RBCs bearing the Wr^b^, En(a-), or Dantu phenotype, their expressions of AE1 and GPA/GPB were mutually affected [26,42,43,44]. Third, Pang et al. showed that the glycosylation time courses of GPA/GPB and AE1 are linked to their intracellular trafficking patterns [23]. The direct interaction between AE1 and GPA conceivably facilitates oligomerization and forward trafficking of the AE1-associated protein complexes that also comprise GPB, Rh/RhAG, and several other erythroid proteins. AE1-associated complexes are considered AE1-central metabolic hubs on the RBC surface [11,29,32,40,45,46,47,48].

Pang et al. found that the ER processing of endogenous GPA in the human K562 erythroleukemic cell line is very efficient and takes only 5–10 min for the glycans of GPA to be modified from high mannose forms to complex ones. Endogenous GPA in K562 cells bears complex carbohydrates, and primarily resides in the Golgi apparatus and on the plasma membrane. Heterologously expressed AE1 bears a high content of mannose and is much retained in the ER of K562 cells [23]. A high content of mannose in a glycoprotein indicates that this protein is most likely to be retained in the endoplasmic reticulum. However, AE1 can traffic to the cell surface without the aid of GPA, and its expression does not involve major ER chaperones, such as calreticulin (CRT) or calnexin (CNX) [49]. In our study, using HEK-293 cells as the heterologous expression system, AE1 was expressed substantially in the plasma membrane (Figure 3 and Figure 4). GPA can further increase the surface expression of other O-glycosylated erythroid membrane proteins [17], as well as the surface expression of AE1 in several heterologous expression systems and in human RBCs (Figure 2, Figure 3 and Figure 4) [11,12,13,26,33,50]. Therefore, GPA conceivably could enhance AE1 surface expression through their oligomerization/glycosylation/maturation while trafficking to the plasma membrane.

AE1 is a versatile and functionally important protein that facilitates major physiologic functions (blood CO_2_ metabolism and acid-base homeostasis), RBC structural stability, as well as red cell processing of nitric oxide (NO) [20,21,51,52,53,54,55,56]. Glycophorin A, glycophorin B and variants, such as GP.Mur, are involved in AE1 complex formation and surface expression to different extents [54]. Their trafficking signals, such as TMD and R18 identified in this study, conceivably may play a critical role in modulating or fine-tuning erythroid AE1 complex expression and functions.

## Figures and Tables

**Figure 1 cells-11-03512-f001:**
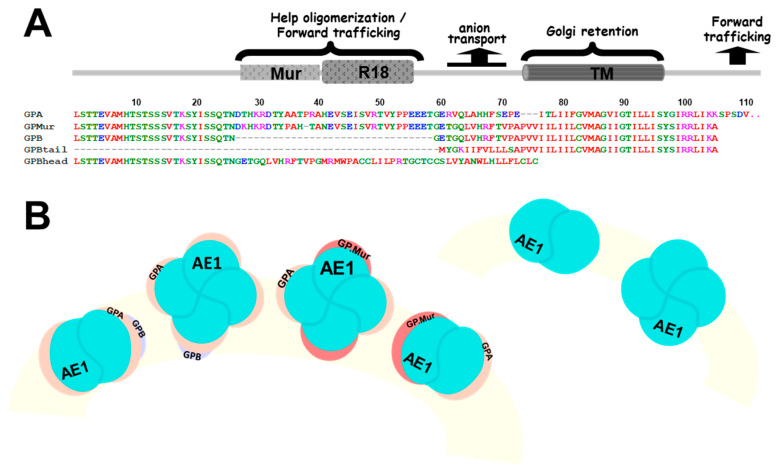
Structural-functional correlates for GPA/GPB and their variants. (**A**) Sequence alignments of GPA, GPB, and variants—GP.Mur, GPBtail, and GPBhead. Most C-terminal, cytoplasmic residues of GPA (25 amino acids) were omitted. The cytoplasmic region of GPA has been reported to support AE1 forward trafficking [33]. The pre-TMD of GPA (residues 61–70) supports the anion transport activity of AE1 [12,33]. GP.Mur is distinguished from GPB by an additional O-glycopeptide that presents the Mur and R18 epitopes; this additional extracellular sequence supports biosynthesis and surface expression of AE1 complexes [11]. This study identified an ER/Golgi retention signal encoded in the homologous glycophorin TM domain, which could be counteracted by the forward-trafficking signals encoded in cytoplasmic GPA and the R18 epitope, or by AE1 coexpression. (**B**) The cartoon depicts GPA/GPB and their variant GP.Mur in dimeric and tetrameric AE1 complexes (top view on the RBC membrane). Though AE1 can form dimers and tetramers and traffic to the plasma membrane without the assistance of GPA, GPA (yellow symbols) and AE1 (blue-green balls) generally interact in a one-to-one fashion. GPB (red symbols) is also present in erythroid AE1-associated complexes, and has been hypothesized to interact with AE1 indirectly through GPA. The copy numbers of GP.Mur (orange symbols) and GPB are about one-fifth the number of GPA on the erythrocyte membrane. Like GPA, GP.Mur directly binds AE1 and promotes AE1 surface expression.

**Figure 2 cells-11-03512-f002:**
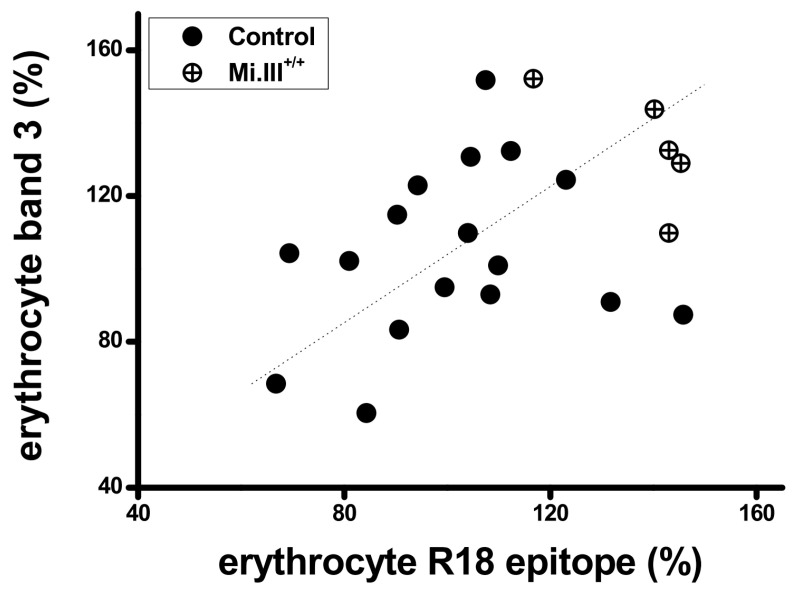
AE1 levels were directly proportional to the GPA-R18 levels on the erythrocyte membrane; *GYP.Mur^+/+^* RBCs generally expressed more AE1 and more R18 epitopes from both GPA and GP.Mur. Fresh RBC samples were immunostained with R18 mAb or with a mix of anti-AE1 mAbs BRIC 6 and BRIC 71 (1:500 dilution each), followed by secondary antibody labeling and flow cytometry. Here, each dot represents the relative levels of AE1 and the R18 epitope of a RBC sample. The geometric means for all non-Miltenberger RBC samples were averaged and set as 100%; all data were expressed in percentage (%). The linearly fitted line showed a rough correlation between the relative levels of AE1 and the R18 epitope on the erythrocyte surface. The data from homologous Mi.III (*GYP.Mur^+/+^*) samples were shown in crossed symbols.

**Figure 3 cells-11-03512-f003:**
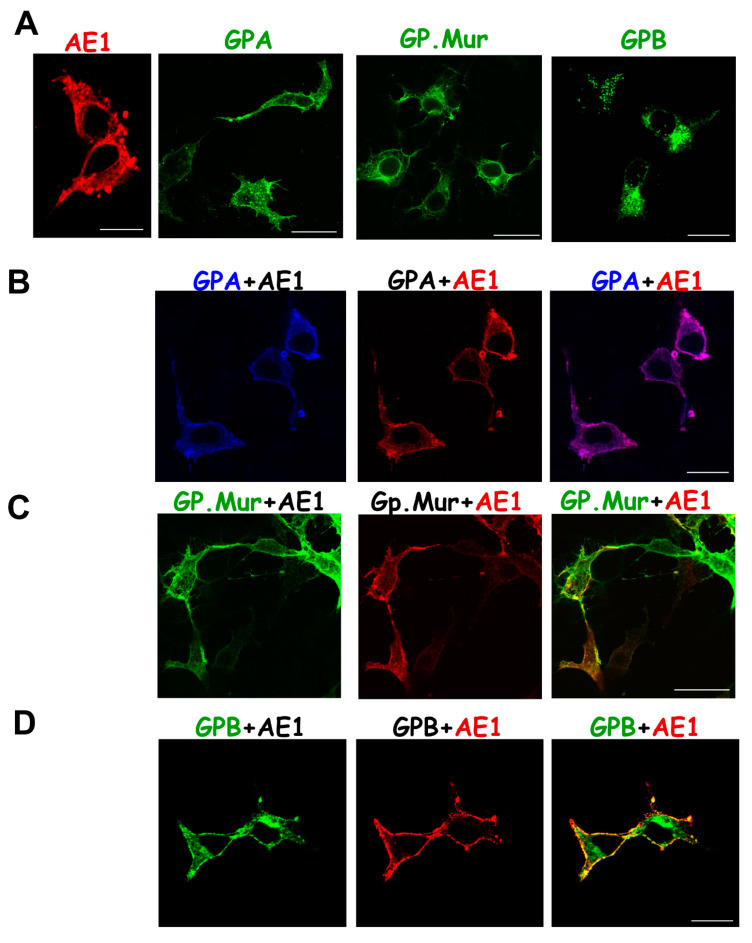
Confocal images revealed subcellular localization of (**A**) singly expressed AE1, GPA, GPMURgfp, and GPBgfp; (**B**–**D**) AE1 coexpressed with GPA, GPMURgfp, or GPBgfp in equimolar ratio; and (**E**) AE1 coexpressed with both GPA and GPBgfp in HEK-293 cells. GPBgfp and GPMURgfp fusion proteins were used for tracking, as there is no GPB- or GP.Mur-specific antibody available for immunolabeling. The transfected cells were fixed and permeabilized for immunostaining of surface and intracellular AE1 and GPA. To visualize AE1 expression, the cells were immunostained with a mix of mouse anti-AE1 mAbs (BRIC 6, BRIC 71, and BRIC170), followed by Alexa Fluor 568 or 647-conjugated anti-mouse antibody. To visualize GPA expression, the transfected cells were immunostained with Alexa Fluor 568-conjugated anti-GPA mAb BRIC163 that targets cytoplasmic GPA (absent in GPB/GP.Mur). In (**E**), HEK-293 cells were transfected with pcAE1, pcGPA and pGPBgfp (their plasmid molar ratio as 1:1:0.2 to mimic the relative levels in human erythrocytes). AE1 and GPA were immunostained as above. Their fluorescence signals were shown in pseudocolor: blue for GPA, red for AE1, and GPBgfp emitting green fluorescence. Scale bar = 25 μm.

**Figure 4 cells-11-03512-f004:**
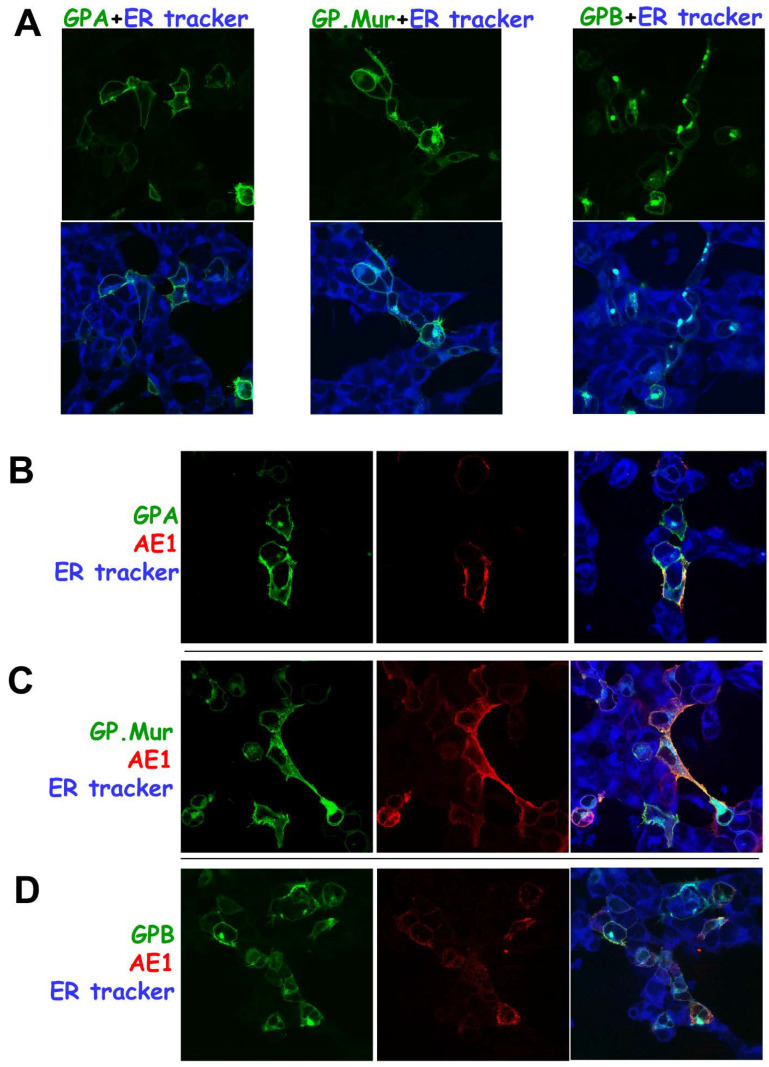
Different from GPA and GP.Mur with substantial surface expression, singly expressed GPB was much retained in the ER but could be driven to the cell surface upon AE1 coexpression. (**A**) TOP: Expression of the fluorescent fusion GPAyfp alone (left), GPMURgfp alone (middle), and GPBgfp alone (right). BOTTOM: Overlay of each glycophorin-fluorescent fusion protein with the blue-fluorescent ER tracker. (**B**) HEK-293 cells cotransfected with equimolar pGPAyfp and pcAE1 were fixed and permeabilized on the second day post-transfection for immunofluorescence labeling of AE1 in red (mixed BRIC 6 + BRIC 71 + BRIC 170 mAb [1:500 dilution each] followed by anti-mouse IgG conjugated Alexa fluor 568 [1:200 dilution]). (**C**) HEK-293 cells were cotransfected with equimolar pGPMURgfp and pcAE1, followed by the same staining protocol as in (**B**). (**D**) HEK-293 cells were cotransfected with equimolar pGPBgfp and pcAE1, followed by the same staining protocol as in (**B**). In (**B**–**D**), the left panel showed expression of the glycophorin-fluorescence fusion protein (in green color) upon AE1 coexpression; the middle panel showed expression of AE1 (in red color) upon glycophorin coexpression; the right panel showed the merged fluorescent signals of the glycophorin fusion (green), AE1 (red) and the ER tracker (blue).

**Figure 5 cells-11-03512-f005:**
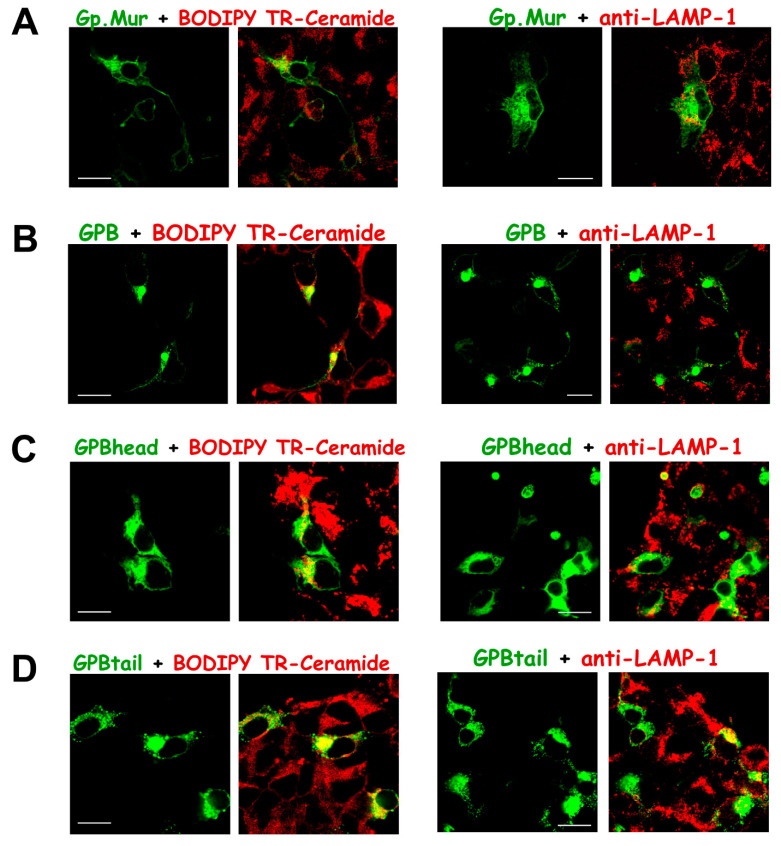
GPB and GPBtail, but not GP.Mur or GPBhead, expressed predominantly in the Golgi apparatus, suggesting a Golgi retention signal embedded in the glycophorin TM domain. HEK-293 cells were transfected with only (**A**) GPMURgfp, (**B**) GPBgfp, (**C**) GPBhead-GFP, or (**D**) GPBtail-GFP. On the second day post-transfection, the cells were labeled with a lipid marker for the Golgi apparatus—BODIPY TR-ceramide (left panel), or with mouse anti-LAMP-1 (a protein marker for lysosomes) following AF568 conjugated anti-mouse antibody (right panel). Scale bar = 25 μm.

**Figure 6 cells-11-03512-f006:**
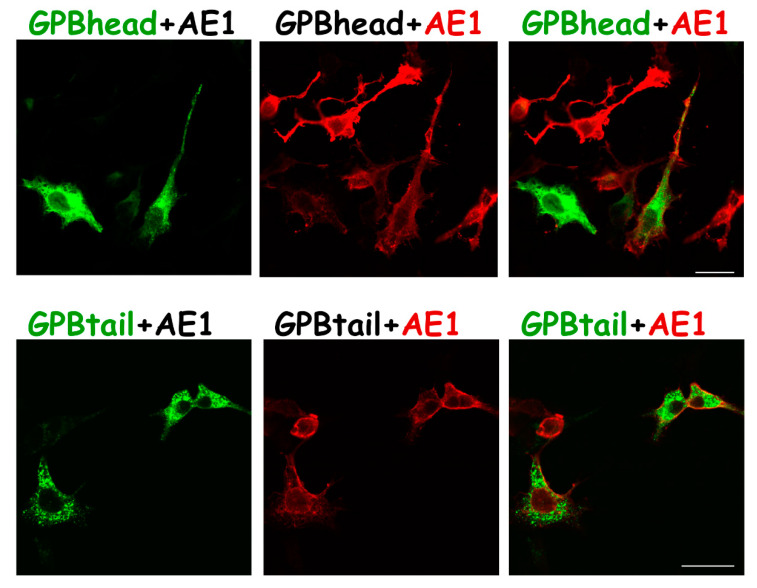
Subcellular localization of GBhead-GFP or GBtail-GFP was not affected by AE1 coexpression, suggesting that there was not even remote interaction between AE1 and the partial GPB structure. (TOP) HEK-293 cells were cotransfected with equal copy number of pGPBhead-GFP and pcAE1 plasmids, followed by immunolabeling with mouse anti-AE1 and then AF568 conjugated anti-mouse mAb. Similar to the GPBhead-GFP expression alone (Figure 5C), the distribution of GPBhead-GFP in AE1-coexpressed cells remained largely cytosolic and did not show any association with AE1 on the cell membrane. (BOTTOM) HEK-293 cells were cotransfected with equimolar pcAE1 and pGPBtail-GFP plasmids, followed by the same immunolabeling protocol to track AE1, as described above. Scale bar = 25 μm.

**Figure 7 cells-11-03512-f007:**
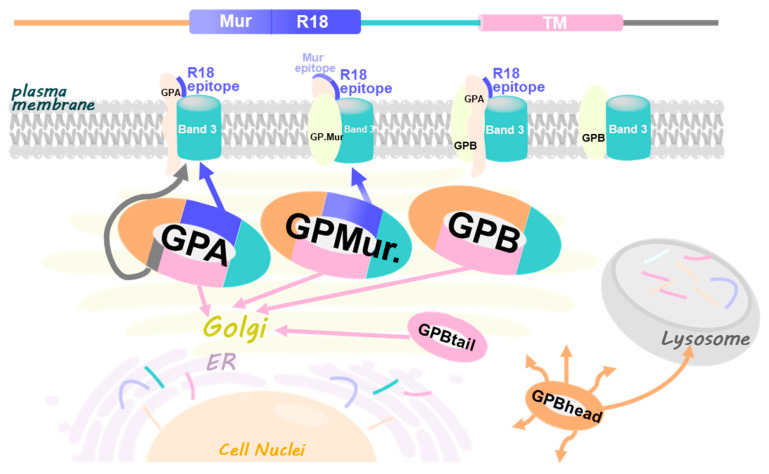
A working model illustrates different trafficking signals encoded in the glycophorin sequence and manifested in glycophorin-AE1 interaction. (TOP) The common sequence of GPA/GPB/GP.Mur and variants are divided in color-coded segments (from left to right, or the N- to the C-terminus): the N-terminal sequence (orange), the Mur epitope (light purple), the R18 epitope (purple); the pre-TMD region that interacts with band 3 and facilitates band 3-mediated anion transport (green); the TMD (pink), and the long C-terminal sequence only in GPA (dark gray). The 5 glycophorin variants studied are each presented in a hoop ring with color segments corresponding to the sequence segments described above. GPA contains all the sequence segments but the Mur epitope. GPB lacks the Mur and the R18 epitopes, as well as the long C-terminus unique to GPA. GP.Mur, evolved from GPB, contains the Mur and the R18 antigens, and also lacks the long C-terminus. GPBtail mainly expresses the TMD. GPBhead mainly expresses the N-terminal glycophorin sequence. The lack of the TMD in GPBhead makes it a diffusive, cytosolic protein destined for the lysosomes. Glycophorin transmembrane is essential for protein retention in the ER and the Golgi apparatus. The glycophorin-R18 epitope (purple) is adjacent to the pre-transmembrane region (green) that includes the Wr^b^ motif (arisen from the AE1-GPA/GP.Mur interaction interface) and functions to assist band 3-mediated anion transport [12,24,26,33,40]. Thus, R18-mediated forward trafficking in GPA and GP.Mur may involve protein–protein interaction with band 3 and co-migration to the cell surface. The long C-terminal sequence of GPA also encodes a forward trafficking signal [33].

## Data Availability

The data presented in this study are available on request from the corresponding author.

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
