# Peer review of "A Balance between Transmembrane-Mediated ER/Golgi Retention and Forward Trafficking Signals in Glycophorin-Anion Exchanger-1 Interaction"

_cells, 2022, doi:10.3390/cells11213512_

Round 1

Reviewer 1 Report

The topic of the paper A Balance between Transmembrane-mediated ER/Golgi Retention and Forward Trafficking Signals in Glycophorin-Anion 3 Exchanger-1 Interaction proposed for publication on Cells fits well with the scopes of the Journal.

The manuscript gives a sight on the two most abundant membrane proteins in human red blood cells, Anion exchanger-1 (AE1; band 3; SLC4A1) and Glycophorin A (GPA) and their relationship.

The article is well presented and supported by clear results.

A minor change is suggested: the discussion as well as the abstract should include a final remark about the biological meaning of the study and its potential applications.

After that,  I consider the paper  suitable for publication.

Author Response

We thank the reviewer for the comments. As suggested, in this revision, we have revised the abstract and the Discussion section to emphasize its biological meaning (highlighted in yellow). We also added a Figure 7 (working model) to explain the findings.

Reviewer 2 Report

I have found this report very interesting. Especially, a role of GPA domain comprising of R18 epitope in formation and forward trafficking of GPA/AE1 komplex. GPB seems to not support the AE1 expression.

My only suggestion is to improove the abstract to be more understandable and  clear not only for  glycophorin's experts.

Author Response

We thank the reviewer for the comments. In this revision, we have rewritten the abstract. For readers not in the field, we have also drawn Figure 7 (a working model) to help explain the findings.
